# Risk averse reproduction numbers improve resurgence detection

**Kris V. Parag** [1] *, **Uri Obolski** [2,3]

**1** MRC Centre for Global Infectious Disease Analysis, Imperial College London, London, United Kingdom, **2** Department of Epidemiology and Preventive Medicine, School of Public Health, Faculty of Medicine, Tel Aviv University, Tel Aviv, Israel, **3** Porter School of the Environment and Earth Sciences, Faculty of Exact Sciences, Tel Aviv University, Tel Aviv, Israel

* k.parag@imperial.ac.uk.

**Data Availability Statement:** We provide open-source software to reproduce all analyses at https://github.com/kpzoo/risk-averse-R-numbers. While the main code for generating the figures in this text is written in MATLAB, we also include

## Abstract

The *effective reproduction number R* is a prominent statistic for inferring the transmissibility of infectious diseases and effectiveness of interventions. *R* purportedly provides an easy-to-interpret threshold for deducing whether an epidemic will grow (*R*>1) or decline (*R*<1). We posit that this interpretation can be misleading and statistically overconfident when applied to infections accumulated from groups featuring heterogeneous dynamics. These groups may be delineated by geography, infectiousness or sociodemographic factors. In these settings, *R* implicitly weights the dynamics of the groups by their number of circulating infections. We find that this weighting can cause delayed detection of outbreak resurgence and premature signalling of epidemic control because it underrepresents the risks from highly transmissible groups. Applying *E-optimal* experimental design theory, we develop a weighting algorithm to minimise these issues, yielding the *risk averse reproduction number E*. Using simulations, analytic approaches and real-world COVID-19 data stratified at the city and district level, we show that *E* meaningfully summarises transmission dynamics across groups, balancing bias from the averaging underlying *R* with variance from directly using local group estimates. An *E*>1 generates timely resurgence signals (upweighting risky groups), while an *E*<1 ensures local outbreaks are under control. We propose *E* as an alternative to *R* for informing policy and assessing transmissibility at large scales (e.g., state-wide or nationally), where *R* is commonly computed but well-mixed or homogeneity assumptions break down.

## Author summary

How can we meaningfully summarise the transmission dynamics of an infectious disease? This question, although fundamental to epidemiology and crucial for informing the design and implementation of interventions (e.g., quarantines), is still not resolved. Current practice is to estimate the *effective reproduction number R*, which counts the average number of new infections generated per past infection, at large scales (e.g., nationally). An estimated *R*>1 signals epidemic growth. While *R* is easily interpreted and computed in real time, it averages infections across diverse locations or socio-demographic groups that

functions in R to compute E on user-defined datasets.

**Funding:** KVP acknowledges funding from the MRC Centre for Global Infectious Disease Analysis (reference MR/R015600/1), jointly funded by the UK Medical Research Council (MRC) and the UK Foreign, Commonwealth & Development Office (FCDO), under the MRC/FCDO Concordat agreement and is also part of the EDCTP2 programme supported by the European Union. UO was supported by a grant from Tel Aviv University Center for AI and Data Science 417 (TAD) in collaboration with Google, as part of the initiative of AI and DS for social good. The funders had no role in study design, data collection and analysis, decision to publish, or manuscript preparation. For the purpose of open access, the author has applied a 'Creative Commons Attribution' (CC BY) licence to any Author Accepted Manuscript version arising from this submission.

**Competing interests:** We declare no competing interests.

likely possess different transmission dynamics. We prove that this averaging in $R$ reduces sensitivity to resurgence, making $R>1$ slow to reflect realistic epidemic growth. This delay can substantially misinform policymakers and impede interventions. We apply optimal design theory to derive the *risk averse reproduction number E* as an alternative summary of diverse transmission dynamics. Using mathematical arguments, simulations and empirical COVID-19 datasets, we show that $E>1$ is an improved threshold for resurgence, providing timelier signals for informing policy or interventions and better uncertainty quantification. Further, $E$ maintains the computability and interpretability of $R$. We propose $E$ as meaningful statistic at large scales, where the averaging within $R$ likely misrepresents the diversity of transmission dynamics.

## Introduction

The *effective reproduction number R*, summarises the time-varying transmissibility or spread of an infectious disease by the average number of secondary infections that it generates per effective primary infection [1]. A value of $R$ above or below 1 is interpreted as a threshold [2] signifying, respectively, that the epidemic is growing (spread is supercritical) or under control (subcritical). This interpretation is widely used, and estimates of $R$ computed at various scales, ranging from e.g., the district to country level when scale is defined spatially, have yielded valuable insights into the transmission dynamics of numerous pathogens including pandemic influenza, malaria, Ebola virus and SARS-CoV-2 [3–5]. During outbreaks, $R$ is monitored and reported in real time to assess the effectiveness of interventions [2,6], signal the emergence of pathogenic variants [7], estimate the probability of sustained outbreaks [8], increase public awareness [9] and inform public health policymaking [10].

The benefits of $R$ mainly stem from two properties: it is easily interpretable as a threshold parameter, and it is easily computable in real time, requiring only routine surveillance data such as epidemic curves of cases [11,12]. However, these must be balanced against its core limitation– $R$ is commonly derived under a well-mixed assumption in which individuals are homogeneous and have equal probabilities of encountering one another. Generalising this assumption to account for the fact that realistic contact rates are heterogeneous, leading to assortative and preferential mixing [13], often necessitates some loss in either interpretability or computability [14]. As examples, we discuss agent-based [15,16] network [17,18] and compartmental [19,20] models. These are common and overlapping approaches to including heterogeneity in transmissibility that can possess complementary characteristics.

Agent-based and network models explicitly describe individual-level epidemic dynamics [17]. They incorporate structures and rules that define heterogeneous patterns of connectivity or mixing among individuals or larger groups such as communities. Although the distinction can be arbitrary, network models may focus more on capturing interaction patterns while agent-based models may emphasise individual-level behaviour. Both approaches require extensive high-resolution data to parametrise connectivity and can become computationally intractable [21]. Connectivity, and hence transmission heterogeneity, may be directly fit from contact tracing data or approximated from large-scale mobility and survey datasets [22–24]. Further, these models can allow us to probabilistically infer connectivity by comparing their outputs to observed data, leading to stratified contact matrices that define the mixing levels among groups [24–26]. However, there are questions about the interpretability and even existence of reproduction numbers for agent-based and network models [27–29].

In contrast, compartmental models, which class or group individuals by their epidemiological state (e.g., susceptible or infected) and assume homogeneous transmission in their simplest formulation, have explicitly defined reproduction numbers [19]. Compartmental models can be extended to depict more complex interactions by including additional classes, which may be informed by contact matrices derived from empirical data or from previous agent-based or network models. This allows compartmental approaches to represent heterogeneous transmission without incurring as much computational overhead or intractability [19,20]. Unfortunately, these additional classes also require more epidemiological rate or distribution data, limiting real-time usage and reducing the interpretability of outputs.

An alternative that can reduce the severity of these complexity-interpretability trade-offs is to model the epidemic as a metapopulation or multitype process, in which local scales are well-mixed but diverse [30–32]. Global scales then enforce structural heterogeneity [14] and we compose the overall $R$ as a weighted function of the local effective reproduction numbers of each group, denoted $R_j$ for group $j$ [33]. We use renewal or autoregressive processes (see Methods) to describe transmission at local scales. These model relationships among routine infection incidence data and the $R_j$ directly and so minimise complexity and computational expense. We consider spatial scales, though analyses often apply to sociodemographic and other types of heterogeneity equally. In our context, local scales may represent regions and the global scale a country composed of those regions. While metapopulation approaches have allowed more informative generalisations of $R$ (e.g., using next generation matrices) [19,34], two key questions remain understudied and form the focus of this paper.

First, can we derive an alternative global statistic that better captures the salient dynamics of local scales than $R$? Standard formulations of $R$ lose sensitivity to key events such as local resurgences [33], defined as a sustained $R_j>1$, because these usually initialise in groups with small incidence, which are down-weighted by $R$. Second, can we optimally trade-off the uncertainty among estimates at local and global scales? In estimating the dynamics of local groups, we reduce the data informing each estimate and are subject to bias-variance trade-offs. If we assess local resurgence using individual $R_j$ estimates, false positives arising from increased uncertainty are more likely and it is unclear how to combine individual estimates to describe overall transmissibility. However, if we neglect local heterogeneities and rely on $R$, we may be statistically overconfident in our estimates of transmissibility.

Using experimental design theory [35], which provides a framework for optimising and fusing estimates of $R_j$ according to cost functions of interest, we derive the *risk averse reproduction number*, $E$. We prove that $E$ upweights groups with likely resurging dynamics because it optimises a cost function that results in the maximum variance among the $R_j$ estimates being minimised. $E$ solves what is known as an E-optimal design problem (see Methods) [35]. Hence $E$ is risk averse and formally trades off bias from $R$ with variance from each $R_j$. We demonstrate, via analytic arguments, simulations and investigation of empirical COVID-19 datasets from Israel, Norway, New Zealand, states within the USA and regions of the UK, that $E$ achieves more meaningful consensus across local-scale dynamics than $R$, improving uncertainty quantification and resurgence detection without losing either interpretability or computability. Given its transparent, design-optimal properties, we believe $E$ can help inform policy at large scales (e.g., state or nationwide) where well-mixed assumptions are invalid.

## Results

### Pitfalls of standard reproduction numbers

We consider the transmissibility of a pathogen at two scales: a local scale, in which we may model a well-mixed population (e.g., within specific geographic or sociodemographic groups)

and a global scale, which integrates the dynamics of $p \geq 1$ local groups. Our local scale, for example, may refer to spread at a district level, whereas the corresponding global scale is countrywide and covers all districts that compose that country. We assume that subdivision into local groups is based on prior knowledge and logistical constraints. We denote the time-varying reproduction number within local group $j$ as $R_j$ and model the dynamics in this group with a Poisson (Pois) renewal model (see Methods) [11] as on the left side of **Eq (1)**.

$$I_j \sim \mathbf{Pois}(R_j \Lambda_j) \text{ for } 1 \leq j \leq p, \qquad \sum_{j=1}^{p} I_j \sim \mathbf{Pois}(\sum_{j=1}^{p} R_j \Lambda_j) \tag{1}$$

This is a widely used framework for modelling time-varying transmissibility [10], with $I_j$ as the new infections and $\Lambda_j$ as the total infectiousness within group $j$. $\Lambda_j$ measures the circulating (active) infections as a weighted sum of past infections in group $j$, with weights set by the generation time distribution of group $j$. We allow this distribution, which describes the times between primary and secondary infections, to vary among the groups [1]. All variables are functions of time e.g., $I_j$ is explicitly $I_j(t)$, but we disregard time indices to simplify notation. An outbreak in group $j$ is growing or controlled if the sign of $R_j - 1$ is, respectively, positive or negative. If a positive sign is sustained, we define group $j$ as being resurgent [33].

We do not directly model inter-group reproduction numbers (e.g., reproduction numbers for infections arising from individuals in group $j$ emigrating into group $i \neq j$). These additional parameters are rarely identifiable from routine surveillance data. However, we can include their impact by distinguishing local from imported infections without altering our methodology [36]. We describe how our formulation includes importations or introductions and implicitly accounts for interconnectivity in later sections (see **Eq (4)** and the Methods). Moreover, **Eq (1)** is widely used in practice [37] to compute $R$ (see below) and we aim to derive statistics that are comparable to $R$. We find, when applied to empirical data (see later COVID-19 case studies), that our simple but completely identifiable statistics work well.

This renewal model approach is also commonly applied over global scales (e.g., to compute national reproduction numbers during the COVID-19 pandemic [37]) by summing infections from every constituent group. This amounts to a well-mixed assumption at this global scale. If we define $I \triangleq \sum_{j=1}^{p} I_j$ and $\Lambda \triangleq \sum_{j=1}^{p} \Lambda_j$ as the new infections and total infectiousness on this global scale, then the transmission model used is $I \sim \mathbf{Pois}(R\Lambda)$, with $R$ as the effective reproduction number on that scale. However, if we instead develop a global model from our local models, we obtain the right side of **Eq (1)**. This simple observation has an important ramification–by assuming that this single $R$ summarises global scale dynamics, we make an implicit judgment about the relative importance of the dynamics in different local groups. We can expose this judgment by simply equating both global models to get **Eq (2)**.

$$R = \sum_{j=1}^{p} w_j R_j, \qquad w_j = \Lambda_j \left(\sum_{i=1}^{p} \Lambda_i\right)^{-1} \tag{2}$$

We see that group $j$ is assigned a weight $w_j$, which is the ratio of active infections in group $j$ to the total active infections at the global scale. Note that $0 \leq w_j \leq 1$ and $\sum_{j=1}^{p} w_j = 1$. This weighting has two key consequences. First, groups with outsized infection loads dominate $R$. This means a group with a large $\Lambda_j$ and small $R_j < 1$, can mask potentially important resurgent groups, which likely possess small $\Lambda_j$ and $R_j > 1$, until those groups generate an appreciable number of infections [33]. Consequently, using $R$ may lead to lagging indicators of resurgence i.e., late warnings. The alternative–to scrutinise each local region for signs of concentrated upticks in $R_j$–may also be suboptimal. Higher stochasticity is expected from data at smaller scales, potentially causing false positive resurgence alarms. This is similar to the classic bias-variance trade-off commonly encountered in statistical modelling [38].

Second, $R$ is only fully representative of local dynamics at two boundary conditions–when the $R_j$ are highly similar and when there is only one active group (i.e., effectively $p = 1$). The latter case is trivial, while the former is unlikely because epidemics commonly traverse connected regions in waves [39] and different groups often possess heterogeneous contact patterns and risks of infection. These all result in diverse $R_j$ time series and desynchronised epidemic curves [40]. Hence, we argue that this commonly estimated $R$ [9,10] may neither be sufficient nor representative for communicating overall transmission risks or informing policymaking. We bolster our argument by showing that $R$ is also statistically overconfident as a summary statistic i.e., its estimates have underestimated variance.

We analyse the properties of $R$ by computing maximum likelihood estimates (MLEs) and Fisher information (FI) values. The left side of **Eq (3)** defines $\hat{R}$, the MLE of $R$, in terms of the MLEs of the $R_j$ of every group. These are $\hat{R}_j = I_j \Lambda_j^{-1}$ [1] under **Eq (1)** (see Methods for derivations). The smallest asymptotic uncertainty around these MLEs (or any consistent $R_j$ estimator) is delineated by the inverse of the FI i.e., larger FI values imply smaller estimate uncertainties [41]. For the renewal models studied here, we know that $\mathbf{FI}[R_j] = \Lambda_j R_j^{-1}$ [42]. When comparing across scales, it is easier to work under the robust or variance stabilising transform $2\sqrt{R_j}$, as it yields $\mathbf{FI}[2\sqrt{R_j}] = \Lambda_j$ (see [42] and Methods for details). We will often switch between the FI of $R_j$ and $2\sqrt{R_j}$ as needed to clarify comparisons, but ultimately will provide main results in the standard $R_j$ formulation.

Substituting our FI expressions into **Eq (2)**, we find the global FI linearly sums the local FI contributions as on the right side of **Eq (3)** and is an increasing function of $p$.

$$\hat{R} = \sum_{j=1}^{p} w_j \hat{R}_j = I \Lambda^{-1}, \qquad \mathbf{FI}[2\sqrt{R}] = \sum_{j=1}^{p} \mathbf{FI}[2\sqrt{R_j}] = \Lambda. \tag{3}$$

Consequently, the uncertainty around $\hat{R}$ is likely to be substantially smaller than that around any $\hat{R}_j$. This formulation underestimates overall uncertainty because the FI acts as a weight that is inversely proportional to the variance of the $R_j$ estimates. Estimates of $R$ are therefore statistically overconfident as measures of global epidemic transmissibility. We demonstrate this point in later sections via the credible interval widths obtained from simulations.

The goal of our study is to design alternatives to $R$ that attain a better consensus across heterogeneous dynamics, with defined properties over diverse locales and without inflated estimate confidence. To achieve these objectives, we must make a principled bias-variance trade-off among signals from $R$ and every $R_j$, deciding how to best emphasise actionable dynamics from local groups without magnifying noise. We apply optimal design theory to develop new consensus reproduction numbers with these tailored properties. As we show in the next section, this involves optimising the weights multiplying every $R_j$ according to cost functions that encode the uncertainty properties and trade-offs that we desire globally.

## D and E optimal reproduction numbers and their properties

The consensus problem of deriving a statistic that is representative of local dynamics can be reframed as an optimal design on the weights mapping the $R_j$ to that statistic, based on a cost function of interest. The uncertainty around estimates of $R_j$, encoded (inversely) by $\mathbf{FI}[R_j] = \Lambda_j R_j^{-1}$, fundamentally relates to key dynamics of the epidemic e.g., resurgence events likely occur at small $\Lambda_j$ and large $R_j$, minimizing $\mathbf{FI}[R_j]$ [33]. Hence, we focus our designs on the Fisher information matrix $\mathbf{FI}_R$ of **Eq (4)**, which summarises the uncertainty from all local $R_j$ estimates. There we replace $\Lambda_j$ with a factor $\alpha_j > 0$ that redistributes the information across

the $p$ groups, subject to its sum being equal to $\Lambda$ (see **Eq (3)**).

$$\mathbf{FI}_R = \begin{bmatrix} \alpha_1 R_1^{-1} & 0 & 0 \\ 0 & \ddots & 0 \\ 0 & 0 & \alpha_p R_p^{-1} \end{bmatrix}, \qquad \text{such that } \sum_{j=1}^{p} \alpha_j = \Lambda. \qquad (4)$$

This formulation facilitates the description of several important scenarios. When $\alpha_j = \Lambda_j$, we recover the standard formulation of $R$ (**Eq (2)**). If we additionally model introductions among groups using probabilities of transporting active infections as in [43], then $\alpha_j$ measures the active infections that are informative about $R_j$ i.e., all infections that are generated in group $j$, including those that are introduced into other groups. This models interconnectedness or inter-group transmissions [36]. If we assume that infections observed in group $j$ are actually a random sample from multiple groups drawn from some multinomial distribution, then $\alpha_j$ corresponds to the fraction of $\Lambda$ assigned by that distribution to group $j$. In the Methods we expand on these points mathematically, showing how **Eq (4)** and the optimal designs below are valid (assuming knowledge of the introductions) when the $p$ groups interact.

Since the $\alpha_j$ are design variables subject to the conservation constraint in **Eq (4)**, we can leverage experimental design theory [35,44] to derive novel consensus statistics to replace the default formulation from **Eq (2)**. We examine $A$, $D$ and $E$-optimal designs, which have standard definitions of how the total uncertainty on our $p$ parameters is optimised. If $p = 2$, this uncertainty can be circumscribed by an ellipse in the space spanned by $R_1$ and $R_2$, and designs have a geometric interpretation as we show in **Fig 1**. $A$-optimal designs minimise the bounding box of the ellipse, while $D$ and $E$-optimal designs minimise its area (or volume, when extending to higher dimensions) and largest chord respectively [44,45]. These designs yield optimal versions of $\Lambda_j$, $\Lambda_j^* = \alpha_j$, computed as shown in **Eq (5)**, where tr[.], det[.] and eig[.]

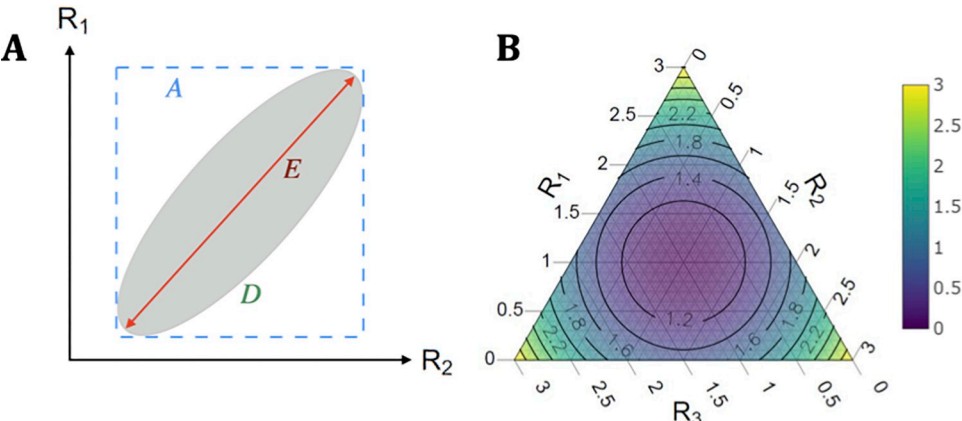

**Fig 1. Illustrations of optimal experimental designs and local reproduction number combinations.** (A) The geometric interpretation of $A$, $D$ and $E$-optimal designs for the $p = 2$ parameter scenario. The overall uncertainty of the parameters is defined by an uncertainty ellipse in the space spanned by possible values of the local reproduction numbers. The ellipse is centred on the MLEs of the parameters and its shape is determined by the inverse of the FI around those estimates. Each design minimises a different characteristic of the ellipse. $A$ minimises the bounding box, $D$ minimises the ellipse area, and $E$ minimises the largest chord (coloured respectively). (B) A ternary plot demonstrating the trajectories of the consensus statistics $D$ and $E$ as a function of different group reproduction numbers $R_j$. These are for $p = 3$ and constrained so $R_1 + R_2 + R_3 = 3$. The colour and contour lines represent $E$ at each combination of $R_j$. We see that $E$ is maximised at the edges, when only one $R_j$ is non-zero. $D$ is at the centre of the triangle, as it is the arithmetic mean of the $R_j$.

indicate the trace, determinant and eigenvalues of their input matrix.

$$\Lambda_j^* | A = \max_{\{\alpha_j\}} \text{tr}[\mathbf{FI_R}], \qquad \Lambda_j^* | D = \max_{\{\alpha_j\}} \text{det}[\mathbf{FI_R}], \qquad \Lambda_j^* | E = \max_{\{\alpha_j\}} \min_j \text{eig}[\mathbf{FI_R}]. \qquad (5)$$

These designs can be done with robust transforms by replacing $\mathbf{FI_R}$ with $\mathbf{FI_{2\sqrt{R}}}$, yielding the diagonal FI matrix $\mathbf{FI_{2\sqrt{R}}} = \text{diag}[\alpha_1, \ldots, \alpha_p]$. We then observe that $\text{tr}[\mathbf{FI_{2\sqrt{R}}}] = \sum_{j=1}^p \alpha_j = \Lambda$. The default allocation of $\Lambda_j^* = \Lambda_j$ is hence, trivially, an $A$-optimal design under this transform. We compute $D$ and $E$-optimal designs without transforms because we want to work with $R_j$ directly. As $\text{det}[\mathbf{FI_R}] = (\prod_{j=1}^p R_j^{-1}) \prod_{j=1}^p \alpha_j$, with the bracketed term as a constant, we find that, subject to our constraint, $\Lambda_j^* | D = p^{-1}\Lambda$. This follows from majorization theory and solutions are adapted from [45,46]. Since $\mathbf{FI_R}$ is diagonal, its eigenvalues are $\alpha_j R_j^{-1}$ and deriving $\Lambda_j^* | E$ equates to solving for $R_1^{-1}\Lambda_1^* = R_2^{-1}\Lambda_2^* = \cdots R_p^{-1}\Lambda_p^*$ (see Methods and [45,46] for derivations). Our $E$ optimal design is then $\Lambda_j^* | E = \Lambda R_j (\sum_{j=1}^p R_j)^{-1}$.

We formulate the new consensus reproduction numbers, $D$ and $E$, by substituting the above optimised $\Lambda_j^*$ into **Eq (2)** to derive **Eq (6)**, which forms our main result. This corresponds to using optimised weights $w_j^* | D = p^{-1}$ and $w_j^* | E = R_j(\sum_{j=1}^p R_j)^{-1}$ in **Eq (2)**. In **Eq (6)** we compute these statistics as convex sums $D = \sum_{j=1}^p (w_j^* | D) R_j$ and $E = \sum_{j=1}^p (w_j^* | E) R_j$.

$$D = \frac{1}{p} \sum_{j=1}^p R_j, \qquad E = \left( \sum_{j=1}^p R_j^2 \right) \left( \sum_{j=1}^p R_j \right)^{-1}. \qquad (6)$$

We refer to $D$ as the *mean reproduction number* because it is the first moment or arithmetic mean of the effective reproduction numbers of each group i.e., it weights the dynamics of each group equally. This construction ensures that the overall uncertainty volume over the estimates of every $R_j$ is minimised. We define $E$ as a *risk averse reproduction number*, and derive it as the ratio of the second to first raw moment of the group reproduction numbers. This is also known as the contraharmonic mean. $E$ weights each group reproduction number by the fraction of the total reproduction number sum attributable to that group. This weighting emphasises groups with large $R_j$, which are considered to be high risk. While $D$ and $E$ do not explicitly include $\Lambda_j$ as in $R$, both are still informed by the active infections because the $\Lambda_j$ (which are proportional to the FI) control the variance of the $R_j$ estimates. This variance or uncertainty propagates into $\hat{D}$ and $\hat{E}$ as in **Eq (6)** (also see Methods).

All three statistics possess important similarities that define them as proxies for reproduction numbers. Because they are all convex sums of the local $R_j$, the value of each statistic lies inside a simplex with vertices at the $R_j$. At the boundary conditions of one dominant group (i.e., essentially $p = 1$) or of highly similar group dynamics (i.e., the $R_j$ are roughly the same over time) this simplex collapses and we find $R = D = E$. Moreover, if all the $R_j = 1$, then $R = D = E = 1$, signifying convergence to the reproduction number threshold. Thus, $D$ and $E$ are alternative strategies to $R$ for combining local reproduction numbers with different properties that may offer benefits when making decisions at large scales. We visualise how these statistics determine our global estimate of transmissibility via the simplex in **Fig 1**. Although we present several reproduction number formulae for comparison, $E$ and its risk averse properties form the main interest of this work.

We refer to $E$ as risk averse because it ensures that the most uncertain $R_j$ is upweighted as compared to the standard formulation of $R$ in **Eq (2)**. This protects against known losses of sensitivity to resurgent dynamics [33] that occur due to averaging across groups, because the FI is expected to be smallest for resurgent groups. As opposed to simply interrogating the individual $R_j$ estimates to identify resurgent groups, $E$ weights those groups, while also accounting

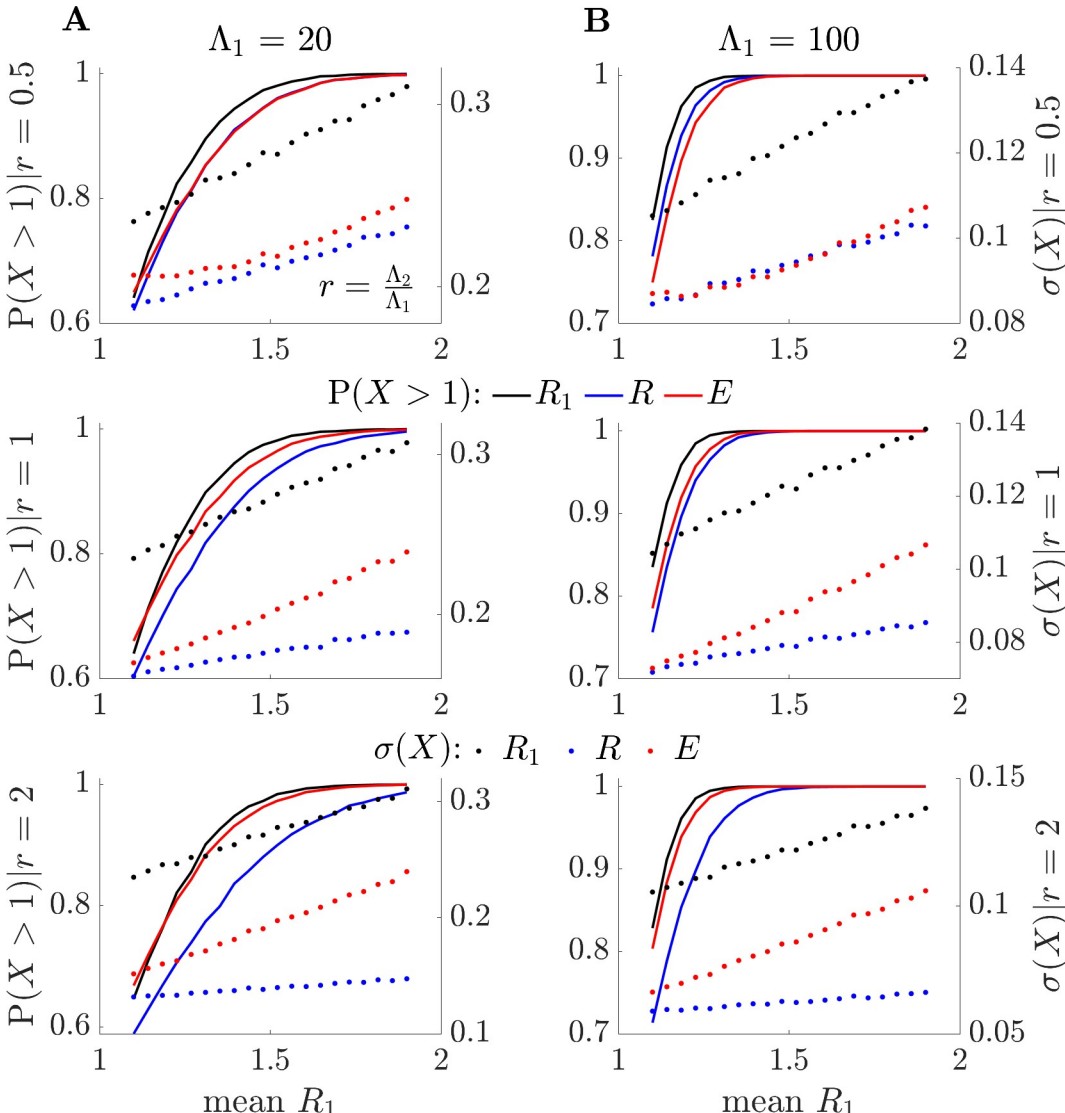

**Fig 2. Relative sensitivity of *E* and *R* to resurgence dynamics.** By sampling from the posterior gamma distributions of [11], we simulate $p = 2$ local groups, varying the values of $R_1$, while keeping $R_2$ at mean value of 1. We plot sensitivities to resurgence of the effective, $R$, and risk averse, $E$, reproduction numbers relative to the maximum local reproduction number $R_1$. These are indicated by the values of $\mathbf{P}(X{>}1)$ for $X = R$, $E$ and $R_1$ (solid blue, red, and black, respectively). (A) and (B) show these resurgence probabilities on left y-axes over a range of mean $R_1$ values for scenarios with small (A) and large (B) numbers of active group 1 infections, $\Lambda_1$. We can assess resurgence sensitivity by how quickly $\mathbf{P}(X{>}1)$ rises and describe the impact of active infections in group 2 using $r = \frac{\Lambda_2}{\Lambda_1}$. We find that $E$ balances the sensitivity between $R_1$ and $R$. The latter loses sensitivity as the active infections in group 2 become larger relative to that of group 1 (i.e., as $r$ increases). This occurs despite group 2 having stable infection counts. We plot the standard deviation of the reproduction number estimates on right-y axes as $\sigma(X)$ for $X = R$, $E$ and $R_1$ (dotted blue, red, and black, respectively). We observe that the local $R_1$ is noisiest (largest uncertainty), while $R$ has the smallest uncertainty (overconfidence). $E$ again, achieves a useful balance.

for the uncertainty in their estimates, to obtain a consensus on overall epidemic transmissibility risk. Consequently, *E* reduces false positives that may occur due to the noise in individual resurgent groups and provides a framework for interpreting situations where some groups may be concurrently resurgent while others are under control. We illustrate these points in **Fig 2** below and explore their ramifications in the next section.

In **Fig 2** we apply the renewal model framework from [11,33], which is based on **Eq (1)** but models estimates of the local reproduction numbers according to the posterior gamma (Gam) distribution $R_j \sim \mathbf{Gam}(I_j, \Lambda_j^{-1})$. This yields a mean estimate of $R_j$ that is equal to the MLE $\hat{R}_j = I_j \Lambda_j^{-1}$ with standard deviation $\boldsymbol{\sigma}(R_j) = I_j^{0.5} \Lambda_j^{-1}$. We consider two local groups ($p = 2$) and compute the probabilities of resurgence $\mathbf{P}(X>1)$ for $X = R_1$, $R$ and $E$ for scenarios that likely represent a resurgence (i.e., small $\Lambda_1$ and increasing $I_1$ with the second group stable at $\Lambda_2 = I_2$). In these cases, $E$ is able to signal resurgence substantially earlier than $R$ but is distinct from simply observing dynamics in the resurging group 1, where $R_1 = \max R_j$. We confirm this by noting that $\boldsymbol{\sigma}(R_1)$ is appreciably larger than $\boldsymbol{\sigma}(E)$, the standard deviation from $E$. Hence responding to $R_1$ is maximally sensitive but also magnifies noise. In contrast $R$ is overconfident, with usually a much smaller $\boldsymbol{\sigma}(R)$.

## Risk averse E is more representative of key transmission dynamics

We test our $D$ and $E$ reproduction numbers against $R$ on epidemics that are simulated from renewal models with Ebola virus generation times from [47] in **Figs 3** and **4**. We also compute max $R_j$ to benchmark how a naive risk averse statistic derived from observing individual groups performs. We consider $p = 3$ groups with various true $R_j$ dynamics (black, dashed) that fluctuate across controlled and resurgent stages. We use the EpiFilter package [48], which applies Bayesian smoothing algorithms, to obtain local estimates (blue) from the incidence curves $I_j$. Similarly, we estimate the overall $R$ (blue) from the total incidence $\sum_{j=1}^{3} I_j$, which is how this statistic is evaluated in practice. We infer $D$ (green) and $E$ (red) by sampling from posterior distributions of local $R_j$ estimates and combining them according to **Eq (6)**. Taking maxima across these local samples gives max $R_j$ (cyan). All estimates include 95% equal tailed Bayesian credible intervals, and we use default EpiFilter settings.

The simulations in **Fig 3** examine abrupt changes in transmissibility. Disease transmission in every group is first controlled. Infections then either resurge ($j = 1,3$) or are driven towards elimination ($j = 2$). Because of its weighting by active infections, $R$ proposes a false, lengthy period of subcritical spread at $t \approx 70$, even though the majority of groups have $R_j>1$. This causes $R$ to be slow to indicate resurgences at $t \approx 100$ and $t \approx 220$. In contrast, $E$ is quick to signal resurgence at $t \approx 220$, without losing the capacity to indicate that the epidemic is under control at $t \approx 140$. $D$ largely interpolates between $R$ and $E$, showing the mean of all the $R_j$ and serves as a null model. As expected from the theory, $R$ is overconfident about its transmissibility estimates, which is apparent from its narrow credible intervals. In contrast, max $R_j$ is very noisy, with considerably larger credible intervals limiting its use.

We further investigate fluctuating but anti-synchronised epidemics ($j = 1,2$) against the backdrop of a much larger monotonically increasing and then decreasing outbreak ($j = 3$) in **Fig 4**. The two out-of-phase groups approximately average to a constant value in both their incident infections and $R_j$. Consequently, we infer a mostly monotonic $R$ and $D$, with $R$ being overconfident in its assessment of transmissibility. In contrast, estimates of $E$ highlight the transmission potential from the fluctuating but smaller epidemics within other groups, while incorporating their uncertainties. It recognises the overall risk across $170 \leq t \leq 270$ posed by groups with fluctuating infections. Additionally, $E$ rapidly signals the transmissibility risk that dominates from $t>270$, which is only indicated by $R$ after a substantial delay. The max $R_j$ statistic is again the most uncertain and prone to false positives.

## Empirical application to COVID-19 across 20 cities in Israel

We compare our consensus statistics with the standard reproduction number on empirical data from the Delta strain outbreak of COVID-19 in Israel across May–December 2021. This

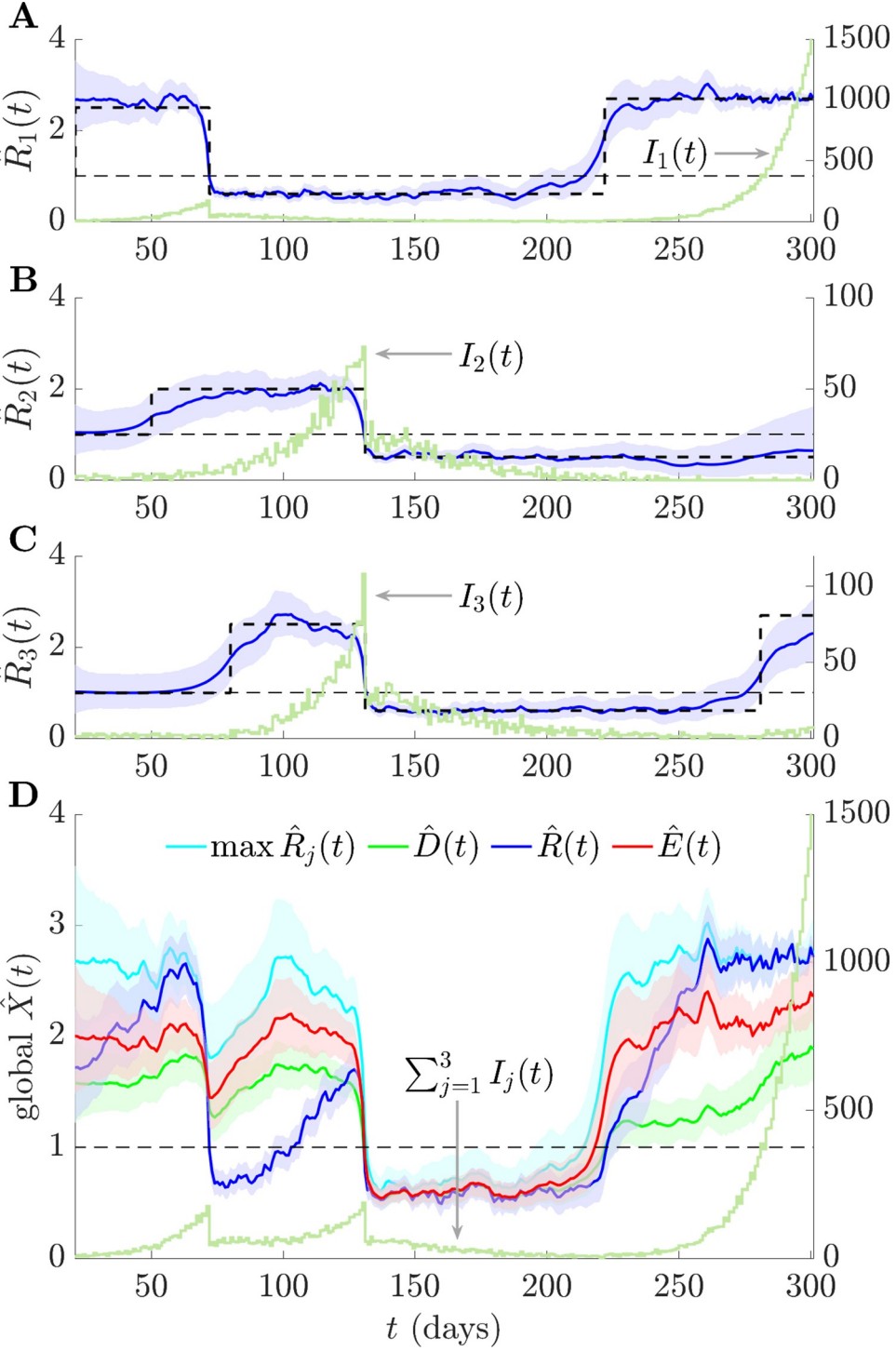

**Fig 3. Consensus statistics for resurging and controlled epidemics.** We simulate local epidemics $I_j(t)$ (dark green) across time $t$ using renewal models with Ebola virus generation times from [47] and true local reproduction numbers with step-changing profiles (dashed black). Estimates of these are in (A)–(C) as $\hat{R}_j(t)$ together with 95% credible intervals (blue curves with shaded regions). (D) provides consensus and summary statistic estimates (also with 95% credible intervals), which we calculate by combining the $\hat{R}_j(t)$. Variations in the standard reproduction number $\hat{R}(t)$ are also reflected in the total incidence $\sum_{j=1}^{3} I_j(t)$. Risk averse $\hat{E}(t)$ and mean $\hat{D}(t)$ reproduction numbers do not signal subcritical spread at $t \approx 70$ (unlike $\hat{R}(t)$) and $\hat{E}(t)$ is most sensitive to resurgence signals. The statistic $\max \hat{R}_j(t)$ is risk averse but magnifies noise. We use EpiFilter [48] to estimate all reproduction numbers.

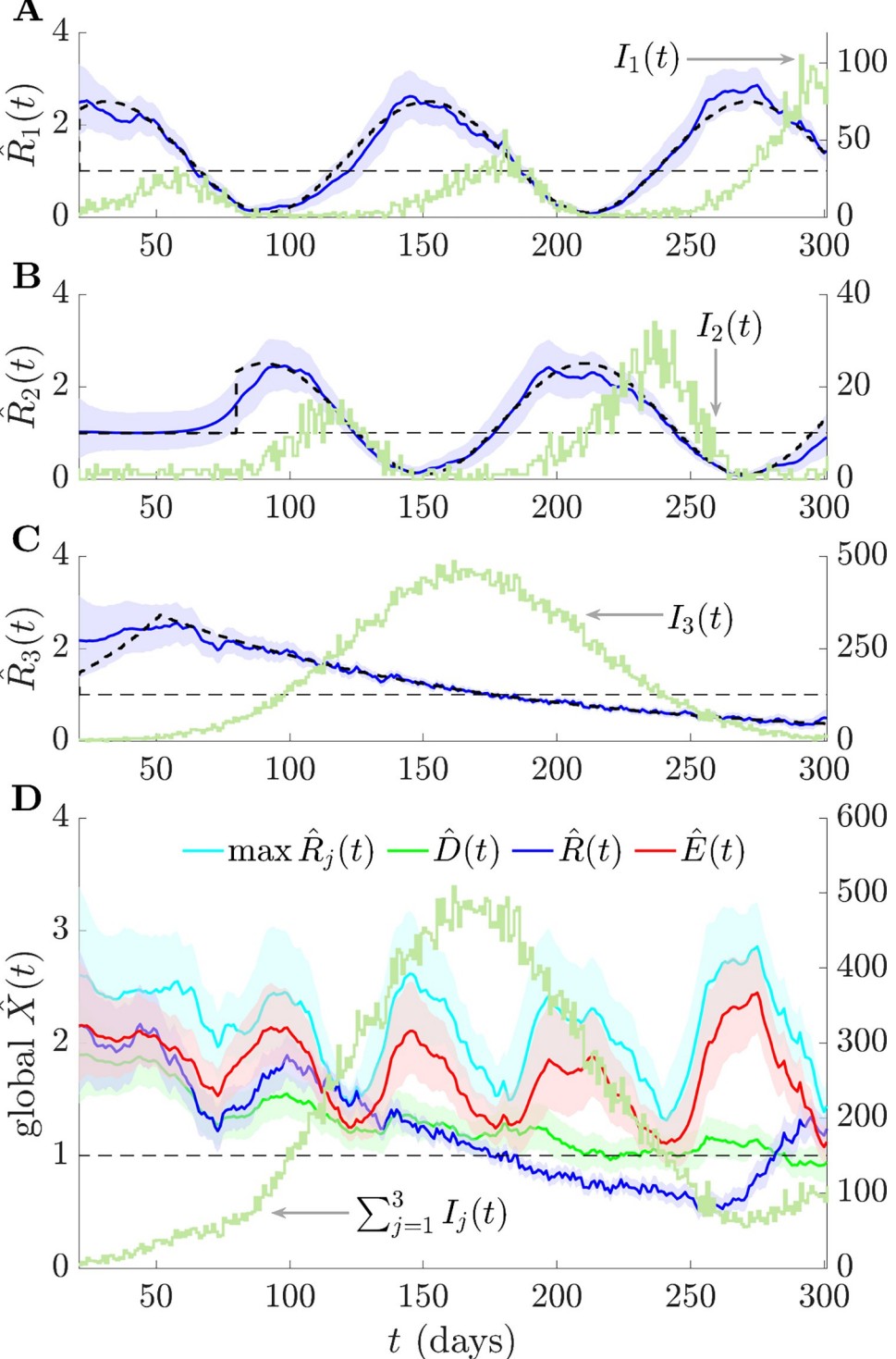

**Fig 4. Consensus statistics for fluctuating and monotonic epidemic dynamics.** We simulate local epidemics $I_j(t)$ over time $t$ from renewal models with Ebola virus generation times as in [47] and true local reproduction numbers with either sinusoidal or monotonically increasing and then decreasing profiles (dashed black). Estimates of these are in (A)–(C) as $\hat{R}_j(t)$ together with 95% credible intervals. (D) plots consensus and summary statistics (also with 95% credible intervals), which we compute by combining those $\hat{R}_j(t)$. Variations in the standard reproduction number $\hat{R}(t)$ are also reflected in the total incidence $\sum_{j=1}^{3} I_j(t)$. Both $\hat{R}(t)$ and the mean $\hat{D}(t)$ reproduction number average over the

fluctuating transmissibility of resurging groups but the risk averse $\hat{E}(t)$ is sensitive to these potentially important signals. Only $\hat{R}(t)$ deems the epidemic to be controlled around $t \approx 200$. The max $\hat{R}_j(t)$ statistic is risk averse but very sensitive to local estimate uncertainties. We use EpiFilter [48] to estimate all reproduction numbers.

dataset provides a convenient case study because daily positive tests results are available from different cities in Israel and both non-pharmaceutical interventions and restrictions were mild during this period. The main intervention deployed, which was highly successful at reducing cases, was the booster vaccine campaign [49,50]. This campaign started July 30 and gradually extended to all ages across August. We examine COVID-19 incidence curves from [51] by date of test for the $p = 20$ cities with the most cases of this wave. These cities account for 49% of the entire caseload in Israel and are plotted in log scale in **Fig 5**.

We estimate the standard, maximum group, mean and risk averse reproduction numbers (labelled as $\hat{X}(t)$) and the probability of resurgence at any time ($\mathbf{P}(\hat{X}(t) > 1)$) as in above sections but with the serial interval distribution in [52], which is consistent with the Israel-specific Delta wave parameters estimated in [50]. We assume case reporting is stable (i.e., any under-reporting is constant) and that serial intervals provide good approximations for the generation times. These assumptions are reasonable given high fidelity surveillance in Israel during this wave and are consistent with the analyses of [49,50]. As we only focus on relative trends, we make no further corrections to the dataset but note that accounting for issues such as testing delays generally cause incidence curves to be back-shifted and increase uncertainty, but necessitate auxiliary data [53,54].

**Fig 5** demonstrates that $D$, $R$ and $E$ all agree that the wave was curbed across the booster period and that the epidemic was controlled. The max $R_j$ is overly sensitive to worst case local dynamics and signals false or early resurgences across October-November. There is substantial disagreement among the statistics prior to the booster campaign. The standard $\hat{R}$ suggests that the wave is under control in May due to decreasing total COVID-19 incidence. However, the risk averse $E$ (and to an extent $D$) highlights potential resurgence and may have contributed evidence to support starting the booster campaign earlier (see **Fig A** of the S1 Appendix for prospective $E$ and $R$ estimates at key timepoints during that period). The max $R_j$ statistic is too susceptible to noise to provide actionable information.

$E$ also better aligns with the emergence of the Delta strain or variant, which is signalled by $\hat{R}$ with substantial delay. Given that counterfactual analyses from [49] showed that the success of the campaign was strongly dependent on the timing of its implementation, this earlier signalling of resurgence could have important ramifications as part of policy response. Using either mean and more conservative statistics (see later figures for visualisation), we find $E$ signals resurgence and supports starting the booster campaign between 2–12 days earlier than $R$ (corresponding to the 8–20 June 2021). The convergence of $D$, $R$ and $E$ across September follows as the epidemic curves of many cities became synchronised and shows that $E$ also recognises periods when dynamics are homogeneous. $E$, via its optimised design, uses the epidemic data to dynamically balance between averaging and emphasising heterogeneous group dynamics. We present a similar analysis on COVID-19 data from Norway that yields qualitatively consistent conclusions in **Fig B** of the S1 Appendix.

## Improved resurgence detection for multiple COVID-19 datasets

We quantify the risk averse behaviour of $E$ in realistic epidemic scenarios by examining its performance on 6 empirical COVID-19 datasets. These include the Israel data above and epidemic curves from districts in Norway (also explored in **Fig B** of the S1 Appendix) and New

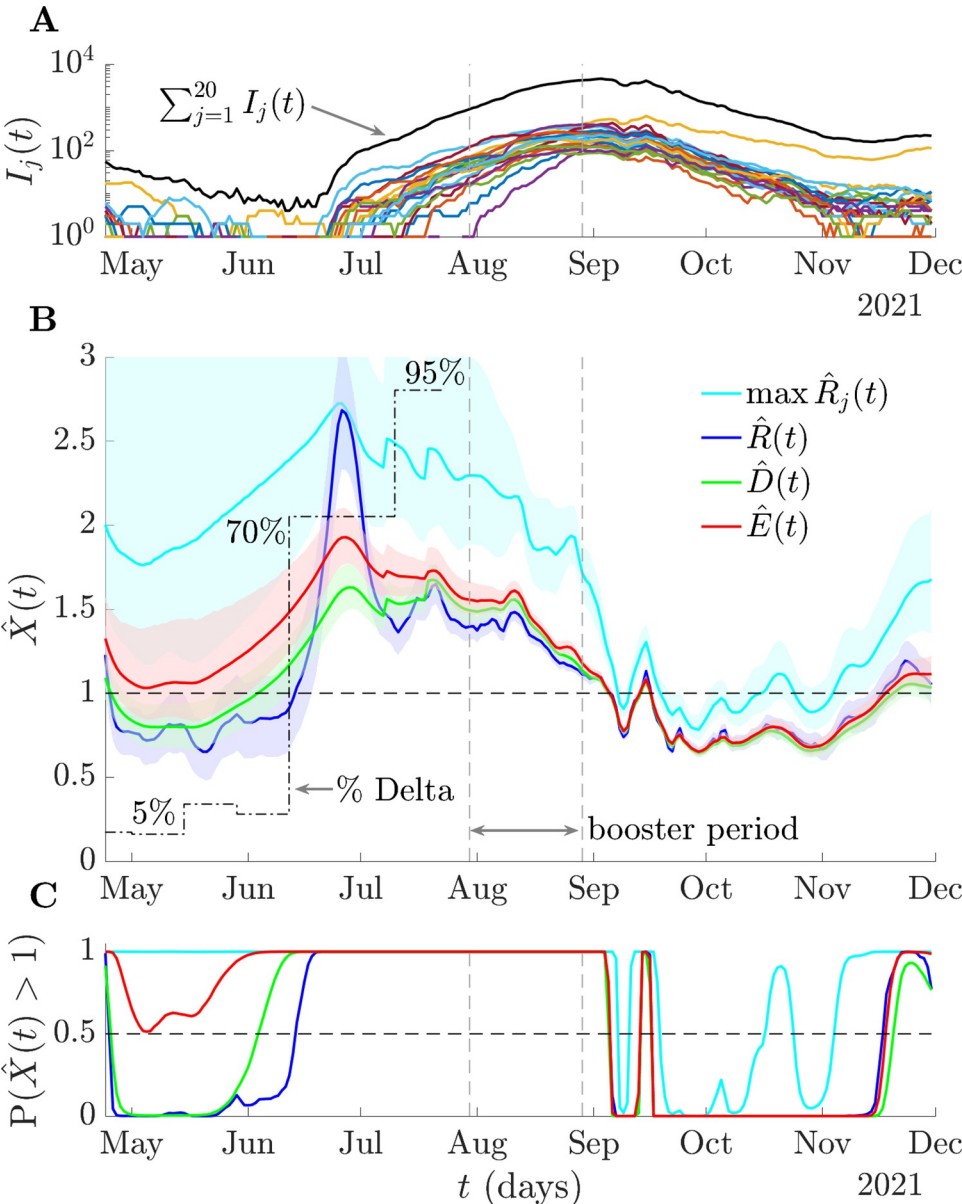

**Fig 5. Risk averse reproduction numbers for COVID-19 in Israel.** We plot the cases by date of positive test (and in log scale) in (A) for $p = 20$ cities in Israel during the Delta wave of COVID-19 from [51]. These constitute 49% of all cases in Israel (summed incidence in black) and have been smoothed with a weekly moving average. We infer the standard, $\hat{R}(t)$, maximum group, max $\hat{R}_j(t)$, mean, $\hat{D}(t)$, and risk averse, $\hat{E}(t)$, reproduction numbers (with 95% credible intervals) using EpiFilter [48] in (B) under the serial interval distribution estimated in [52]. We also plot the proportion of cases attributable to the Delta strain from [50] (black, dot-dashed). We assume perfect reporting and that generation times are well approximated by the serial intervals. (C) integrates the posterior estimates from (B) into resurgence probabilities $\mathbf{P}(\hat{X}(t) > 1)$. While all reproduction numbers indicate effectiveness of the vaccination campaign in curbing spread, $\hat{R}(t)$ is the slowest to signal resurgence across June, at which point the Delta strain has a 70% share in all cases. $\hat{E}(t)$ is more aligned with signalling Delta emergence but avoids the inflated uncertainty of max $\hat{R}_j(t)$.

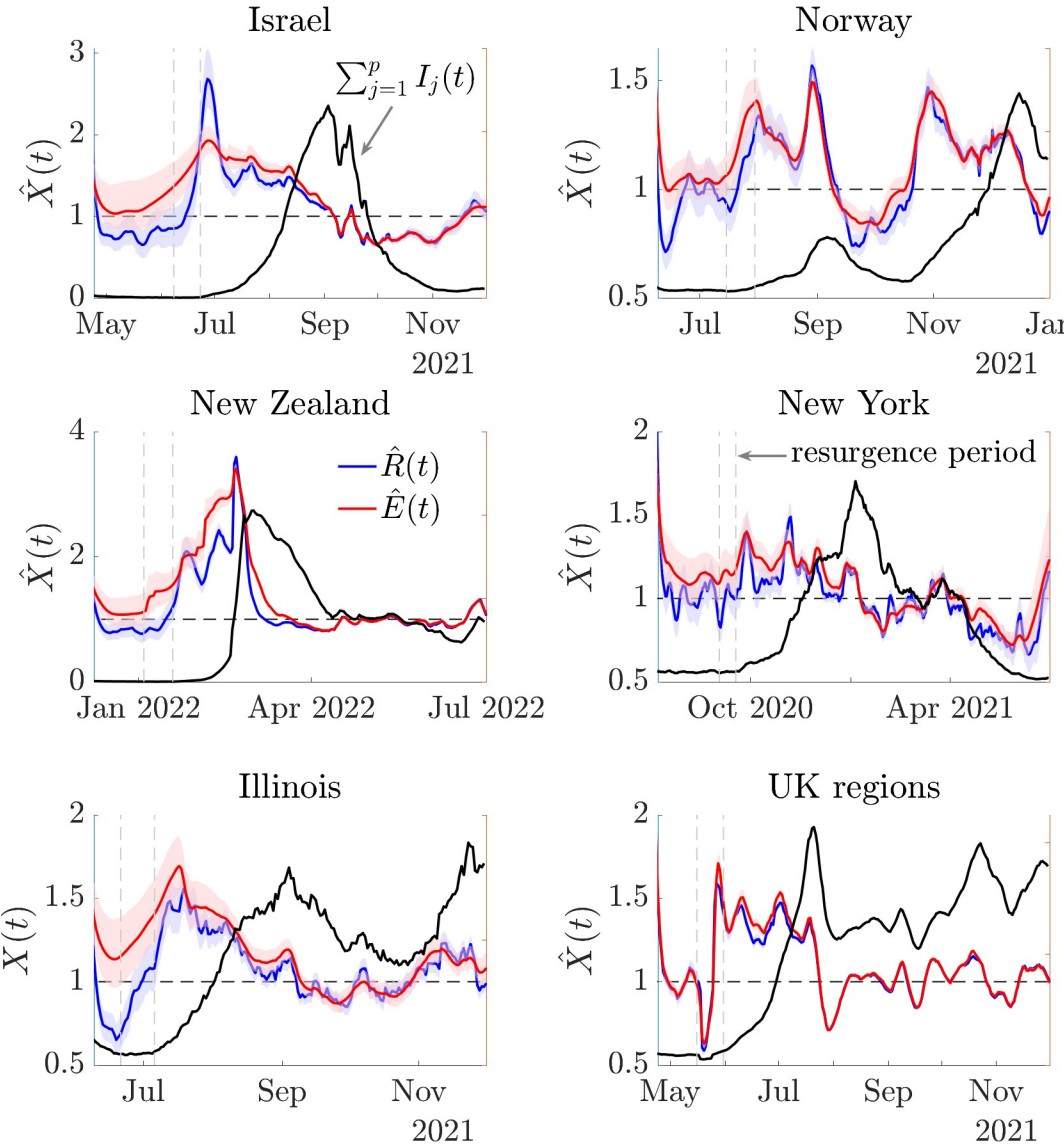

**Fig 6. Transmissibility estimates for COVID-19 in 6 empirical datasets.** We estimate the standard, $\hat{R}(t)$ (blue) and risk averse, $\hat{E}(t)$ (red) reproduction numbers (with 95% credible intervals) using EpiFilter [48] on COVID-19 data describing epidemics in 6 diverse locations (see panel titles). We use the serial interval distribution in [52] and demarcate key periods of resurgence with vertical dashed lines. We investigate these periods in detail in **Fig 7**. Black curves show the shape of the total incidence in the case studies for context. **Fig A** of the S1 Appendix plots the group level incidence, which often feature heterogeneous patterns.

Zealand, regions in the US states of New York and Illinois and local UK authorities. We plot group level and total incidence for all datasets in **Fig A** of the S1 Appendix. We select the top 20 groups by infection counts for each dataset (or all groups when fewer than 20). All curves present cases by date of test (data sourced from [51,55–59]) after weekly smoothing and include resurgences that started locally before propagating. We estimate $R$ (blue) and $E$ (red) for all datasets (retrospectively), under the serial interval distribution from [52] and plot our results in **Fig 6** together with total incidence (black). We indicate some key resurgence periods with vertical lines. We analyse these periods prospectively in **Fig 7**.

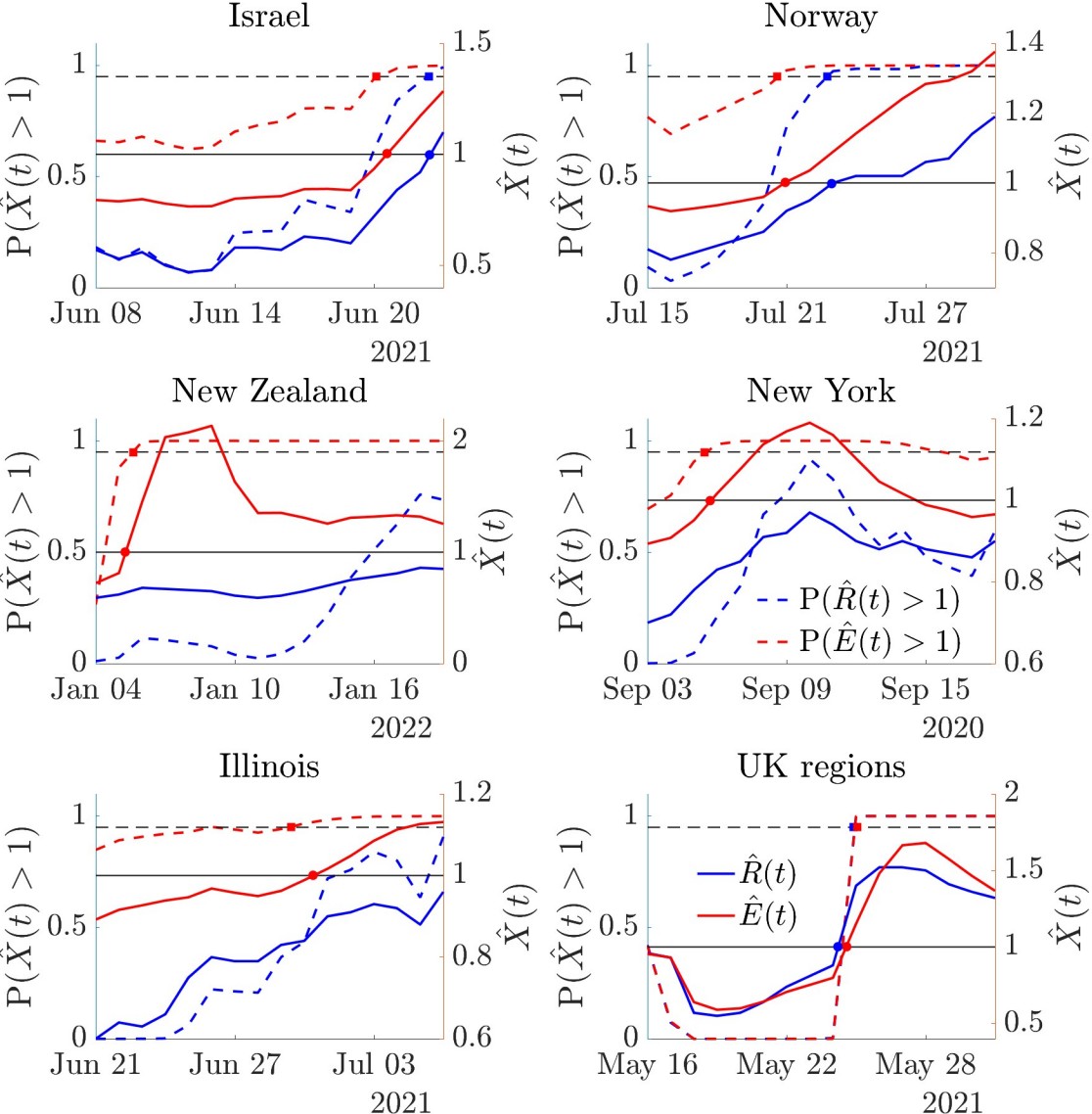

**Fig 7. Resurgence signals from transmissibility estimates for COVID-19 in 6 empirical datasets.** We compute sequential estimates of standard, $\hat{R}(t)$ (blue) and risk averse, $\hat{E}(t)$ (red) reproduction numbers across the periods delimited in **Fig 6** and using the same serial intervals and data described above. These estimates are prospective i.e., at any timepoint they assume that the time series ends at that point (see **Fig A** of the S1 Appendix for more examples). Consequently, these estimates simulate the sequential signals that would have been available about resurgence as incidence data accumulated in real time. Solid lines show lower 95% credible intervals from both $\hat{X}(t)$ relative to a threshold of 1 (solid black, coloured circle intersections). Dashed lines compare $\mathbf{P}(\hat{X}(t) > 1)$ to a probability of 0.95 (dashed black, coloured square intersections). These are conservative metrics.

There we sequentially re-compute our estimates over time and test the resurgence detection ability of both $X = R$ and $X = E$. Estimates at any timepoint in **Fig 7** are informed by data up to that time only, illustrating the resurgence signals we would have inferred if our time series ended at that timepoint. We can decide if reproduction numbers have signalled resurgence in multiple ways. The simplest compares mean estimates $\hat{X}$ to 1 or resurgence probabilities $\mathbf{P}(\hat{X}(t) > 1)$ to 0.5. While we do not show these explicitly, these differences are visible from **Fig 6** and are substantial, often of the order of 1–2 weeks. For example, the relative delay of $R$

in signifying resurgence in the Israel study is 12 days. However, as our estimates possess uncertainty, we may prefer to indicate resurgence more conservatively by finding when the lower limit of the 95% credible interval of $\hat{X}$ crosses 1 or when $\mathbf{P}(\hat{X}(t) > 1) \geq 0.95$ [33].

**Fig 7** plots these estimates. Delays in resurgence detection signals from *R* (blue) relative to those from *E* (red) are visible from the separation of the coloured circles and squares. We find that, except for the null case of the UK regions, where local epidemics are synchronised (see **Fig A** of the S1 Appendix), *E* always provides earlier resurgence signals, confirming its risk averse nature. In the cases of New York, Illinois and New Zealand *R* fails, across the 2-week period analysed, to ever indicate resurgence. For Israel the lag between signals from *R* and *E* is 2 days. The converse occurs if assessing subcritical spread as *E* is slower than *R* to fall below 1 when groups show appreciable heterogeneity (while not explicitly shown we can see this in **Fig 6**). This also confirms the risk averse properties of *E*. We provide general mathematical arguments for why *E* has these properties in the S1 Appendix.

## Discussion

The value of reproduction numbers or similar measures of transmissibility (e.g., growth rates [12]) as statistics for providing actionable information about the state of an epidemic, lies in their ability to accurately identify changepoints between subcritical and supercritical spread [60]. However, the meaning of a changepoint across scales is ambiguous and understudied. For example, if we have *p* local groups, how many need to resurge before we decide the epidemic has become supercritical? Is it a changepoint if these groups resurge at different times? A related question is, if those local groups are heterogeneous, is there any meaning in an overall average [27] such as the standard effective reproduction number *R*? Here we have explored such questions and their implications for describing epidemics at large scales.

We modelled epidemics at two scales: a local scale, over which the well-mixed assumption likely holds, and a global scale, where this assumption is almost surely invalid. Reproduction numbers are commonly computed and reported at global scales. Using this framework, we analysed how changepoints in local reproduction numbers, $R_j$, influence the properties of global statistics. We showed that, due to its weighting of each $R_j$ by the infections circulating in that group, $\Lambda_j (\sum_{j=1}^{p} \Lambda_j)^{-1}$, *R* is generally controlled by the dynamics of the groups with the most extant infections. This causes loss of sensitivity to resurgent changepoints (which may often occur at small $\Lambda_j$) and means that estimates of *R* are usually overconfident or oversmoothed (**Eq (3)**). We attempted to counter these undesirable properties by applying experimental design theory to develop algorithms that optimise the weights on the $R_j$.

We derived a novel reproduction number, *E*, by selecting weightings on the $R_j$ that minimise the maximum uncertainty from $R_j$ estimates. Consequently, *E* upweights more uncertain estimates (often associated with resurgent groups [33]) and incorporates the local circulating infections according to their impact on overall estimate uncertainty. This prevents estimate overconfidence and presents a principled method for combining the local $R_j$ changepoints. An *E*>1 ensures resurging groups are emphasised without being overly sensitive to individual group noise (**Fig 2**), while *E*<1 indicates that groups are under control with high likelihood. Interestingly, *E* weights each $R_j$ by its transmissibility ratio, $R_j (\sum_{j=1}^{p} R_j)^{-1}$, which results in a formula (**Eq (6)**) that seems consistent with that derived from network epidemic models when individuals have heterogeneous contact rates [27].

**Eq (6)**, which is the contraharmonic mean of local reproduction numbers, also suggests that *E* will have general risk averse properties. This follows as contraharmonic means are known to behave as envelope detectors [61] i.e., they detect peaks in waveforms. *E* also has

some resurgence prediction qualities as its weights $R_j(\sum_{j=1}^{p} R_j)^{-1}$ correlate (in rank) with what the weights $\Lambda_j(\sum_{j=1}^{p} \Lambda_j)^{-1}$ in $R$ would converge to if resurgence occurs. We detailed these general properties of $E$ in the S1 Appendix. We further illustrated and validated these properties using multiple simulated and empirical datasets (**Figs 3–7**). There we demonstrated how $E$ provides a better consensus than $R$ of local group dynamics (converging to $R$ when transmission is homogeneous) but is not as vulnerable to noise as the maximum local statistic max $R_j$. We found that the earlier resurgence detection provided by $E$ could be substantial, leading $R$ by up to 2 weeks in several case studies.

This earlier resurgence signalling of $E$ may be important given the sensitivity of the cost and effectiveness of many epidemic control actions to their implementation times [49,62] and the growing evidence supporting earlier but data-driven intervention choices [63,64]. If earlier resurgence signals are ignored, then eventually $\Lambda_j(\sum_{j=1}^{p} \Lambda_j)^{-1}$ will approach $R_j(\sum_{j=1}^{p} R_j)^{-1}$ and $R$ will gradually indicate that supercritical spread has occurred. Due to its risk averse properties, $E$ will also signal subcritical spread at global scales when there is a higher likelihood of groups being under control. This may be more conservative than $R$ but avoids premature relaxations of interventions, which have been correlated with more costly and less effective exit strategies [65]. However, all these benefits depend on socio-political and other factors as reproduction numbers are one of many metrics informing public health decisions.

While $E$ is a promising addition to the suite of infectious disease outbreak statistics, it is not perfect. First, its formulation depends on Poisson noise models (**Eq (1)**). While such models are commonly applied [10], in some cases they may only offer simplistic representations of the stochasticity of epidemics. Although $E$ will likely maintain its risk averse properties due to its contraharmonic formulation, its optimality is unknown for general stochastic descriptions. Second, we defined resurgence and control indicators based on the reproduction number threshold value of 1. This definition is almost universal but other measures (e.g., the early warning signals of [66]) may circumvent some problems that $R$ presents as a statistic for informing decision-making in real time. In some instances, the disease under study may be uncontrollable (e.g., if it possesses long incubation periods and substantial pre-symptomatic spread [67]) and no metric (including $E$) will be able to meaningfully inform health policy.

Third, $E$ requires infection time and incidence data (or proxies [54]) at the resolution of the local scale. While this is becoming the norm with steadily improving surveillance [68], it is not guaranteed and may be scarce for emerging infectious diseases. In this scenario, $R$ is still directly computable from global scale data but the same data resolution limits that prevent inferring $E$ here will also preclude any other finer-scale analysis. Last, the added value of $E$ in informing real-time decision-making depends on the quality of data. Practical biases such as delays in ascertaining cases can hinder timely responses to transmission changepoints [53]. This bottleneck is fundamental and will equally limit $R$ and other statistics. However, in these scenarios, $E$ may still be of use retrospectively.

Overall, we propose $E$ as a consensus statistic that better encapsulates salient dynamics across heterogeneous groups without losing the interpretability or computability of $R$. With late responses to epidemic resurgence often associated with larger epidemic burden [62], increasing interest in early warning signals [66] and reproduction numbers commonly being computed on vast scales in real time [9], the risk averse properties of $E$ may be impactful. Public health policymaking is a complex process combining inputs from diverse data and models spanning epidemiology, economics and behavioural science. Given this complexity, we think that statistics designed with optimal and deliberate properties, such as $E$, can facilitate more transparent and robust data-driven decision-making.

## Methods

### Renewal models and estimation statistics

The renewal model [1] is a popular approach for tracking dynamics of infectious diseases [10]. It describes how the number of new or incident infections at time $t$, $I(t)$, depends on the effective reproduction number at that time, $R(t)$, and the total infectiousness $\Lambda(t)$ as in **Eq (7)**, assuming that infectious individuals mix homogeneously. We commonly define time in days, but the model may be applied at other timescales (e.g., weeks).

$$I(t) \sim \textbf{Pois}(R(t)\Lambda(t)), \qquad \Lambda(t) = \sum_{s=1}^{t-1} \omega(t-s)I(s). \tag{7}$$

Here **Pois** indicates a Poisson noise distribution and $\Lambda(t)$ defines the active or circulating infections as the convolution of earlier infections with the generation time distribution of the disease. This distribution defines the time interval between primary and secondary infections [1] so that $\omega(t-s)$ is the probability of this interval being of length $t-s$. We assume that we have access to good estimates of the generation time distribution and infections [11].

**Eq (7)** has been applied to model many epidemics across a wide range of scales spanning from small communities to entire countries. Its major use has been to facilitate the inference of the time-varying $R(t)$. Fluctuations in estimates of $R(t)$ are frequently associated with interventions or other epidemiologically important events such as the emergence of novel pathogenic variants. We can derive the key statistics of these $R(t)$ estimates from the log-likelihood function of $R(t)$, $\ell$, which follows from the Poisson formulation of **Eq (7)** as in **Eq (8)**. Here $\zeta(t)$ collects all terms that are independent of $R(t)$.

$$\ell = \log \textbf{P}(I(t)|R(t)) = I(t)\log R(t) - R(t)\Lambda(t) + \zeta(t). \tag{8}$$

We can construct the maximum likelihood estimate (MLE) of $R(t)$, denoted $\hat{R}(t)$, by solving $\frac{\partial \ell}{\partial R(t)} = 0$ to obtain the left expression of **Eq (9)**. This estimator is asymptotically unbiased. The (expected) Fisher information (FI), **FI**$[R(t)]$, defines the best achievable precision (i.e., the smallest variance) around the MLE [69], and is computed from **Eq (8)** as $\textbf{E}\left[-\frac{\partial^2 \ell}{\partial R(t)^2}\right]$ [69,70], with $\textbf{E}[.]$ as an expectation over the incidence data. This gives $\frac{\textbf{E}[I(t)]}{R(t)^2}$. Substituting $\textbf{E}[I(t)] = \Lambda(t)R(t)$ from **Eq (7)** produces the expression in the middle of **Eq (9)**.

$$\hat{R}(t) = \frac{I(t)}{\Lambda(t)}, \qquad \textbf{FI}[R(t)] = \frac{\Lambda(t)}{R(t)}, \qquad \textbf{FI}\left[2\sqrt{R(t)}\right] = \Lambda(t). \tag{9}$$

The FI depends on the unknown $R(t)$. We can remove this dependence by applying a robust or variance stabilising transform [45,71]. We can derive this by using the FI change of variables formula as in [42,70]. We consequently obtain the right equation in **Eq (9)**, under the square root transform $2\sqrt{R(t)}$. In the main text we often use this transformed FI to make comparisons clearer but present all key results in the standard $R(t)$ formulation.

While we have outlined the core of renewal model estimation, most practical studies tend to apply Bayesian methodology [10]. Accordingly, we explain the two approaches used in this paper. The first follows [11] and assumes some gamma (**Gam**) conjugate prior distribution over $R(t)$, leading to the posterior estimate of $R(t)$ being described as **Gam**$(I(t), \Lambda(t)^{-1})$ (where we ignore prior hyperparameters). The mean of this posterior is $\hat{R}(t)$ and its variance is the inverse of **FI**$[R(t)]$ evaluated at $\hat{R}(t)$. This formulation holds for group reproduction numbers $R_j(t)$ as well, which have posteriors **Gam**$(I_j(t), \Lambda_j(t)^{-1})$. This methodology has been applied to estimate real-time resurgence probabilities [33]. We use it to generate **Fig 2**.

The second approach is EpiFilter [48], which combines the statistical benefits of two popular $R(t)$ estimation methods–EpiEstim [11] and the Wallinga-Teunis approach [40]–within a Bayesian smoothing algorithm [72] to derive optimal estimates in a minimum mean squared error sense. We apply EpiFilter to obtain all effective reproduction number estimates with their 95% equal-tailed Bayesian credible intervals in **Figs 3–5**. This assumes a random walk prior distribution on $R(t)$ and sequentially computes estimates via forward-backward algorithms [72]. We run EpiFilter at default settings.

It outputs posterior $\mathbf{P}(R_j(t)|I_1^T)$ for group $j$ with $I_1^T$ as the incidence curve $I(t)$: $1{\leq}t{\leq}T$. This provides retrospective analysis of reproduction numbers, using all the information up to present time $T$. If we set $T$ to earlier timepoints we can recover past, real-time estimates that reflect the information available up to that timepoint. We compute these real-time estimates at key timepoints for the Israel case study in **Fig A** of the S1 Appendix. We construct our consensus posterior distributions $\mathbf{P}(X(t)|I_1^T)$, with $X$ as $D$, $R$, max $R_j$, or $E$ (see the next section) by sampling from all $\mathbf{P}(R_j(t)|I_1^T)$ and applying appropriate weightings. Resurgence probabilities are evaluated as $\mathbf{P}(X(t) > 1|I_1^T) = \int_1^\infty \mathbf{P}(X(t)|I_1^T) \, dX$.

## Optimal experimental design and consensus metrics

In the above section we outlined how to model and estimate $R(t)$ across time. However, this assumes that all individuals mix randomly. This rarely occurs and realistic epidemic patterns are better described with hierarchical modelling approaches as in [14]. We investigated such a model at a local and global scale in the main text. There we assumed that $p$ local groups do obey a well-mixed assumption and have local reproduction numbers, $R_j(t)$ for group $j$ that all conform to **Eqs (7–9)**. We additionally modelled a global scale, as in **Eqs (1–3)** that combines the heterogeneous dynamics of the groups. We drop explicit time indices and note that this formulation, which considers weighted means of the $R_j$, requires a $p{\times}p$ FI matrix to describe estimate uncertainty as in **Eq (4)**. We now explain how the consensus statistics, $D$ and $E$, emerge as optimal designs of this matrix.

For convenience, we reproduce the FI matrix $\mathbf{FI}_X$ and the weighting for some reproduction number or consensus statistic $X$ in **Eq (10)**. $X$ can be $D$ or $E$, and we apply a constraint on factors $\alpha_j$ such that $\sum_{j=1}^p \alpha_j = \sum_{j=1}^p \Lambda_j = \Lambda$. When all the $\alpha_j = \Lambda_j$, we obtain the global effective reproduction number, now denoted $R$. The total information of our model is $\Lambda$.

$$\mathbf{FI}_X = \begin{bmatrix} \alpha_1 R_1^{-1} & 0 & 0 \\ 0 & \ddots & 0 \\ 0 & 0 & \alpha_p R_p^{-1} \end{bmatrix}, \qquad X = \sum_{j=1}^p \left( \frac{\alpha_j}{\sum_{j=1}^p \alpha_j} \right) R_j. \tag{10}$$

The mean reproduction number $D$ is derived as the $D$-optimal design of $\mathbf{FI}_X$. This maximises the determinant of this matrix, which is $\left(\prod_{j=1}^p R_j^{-1}\right) \prod_{j=1}^p \alpha_j$. As the first term is independent of the weights, we simply need to maximise $\prod_{j=1}^p \alpha_j$ subject to a constraint on $\sum_{j=1}^p \alpha_j$. This is known as an isoperimetric constraint and is solved when the factors are equalised i.e., $\alpha_j^* = \frac{1}{p}\Lambda$ [35,45]. Substitution of this optimal design leads to an equal weighting $w_j = \frac{\alpha_j^*}{\Lambda} = \frac{1}{p}$ in **Eq (10)** and we get the formulation in **Eq (6)**.

The risk averse reproduction number $E$ is accordingly the solution to the $E$-optimal design of $\mathbf{FI}_X$. This maximises the minimum eigenvalue of $\mathbf{FI}_X$ (i.e., minimises the maximum estimate uncertainty). Because $\mathbf{FI}_X$ is diagonal we must maximise the minimum diagonal element $\alpha_j R_j^{-1}$ subject to the constraint on $\sum_{j=1}^p \alpha_j$. This has a known solution because the objective function

$\min_j \alpha_j R_j^{-1}$ is Schur concave. This objective function is maximised when $\alpha_j^* R_j$ is constant for all $j$ (under our constraint) and yields $\alpha_j^* = \Lambda R_j (\sum_{j=1}^{p} R_j)^{-1}$, which follows from majorization theory. More details can be found in [45,46]. Substituting this optimal design into **Eq (10)** yields the weight $w_j = \frac{\alpha_j^*}{\Lambda} = \frac{R_j}{\sum_{j=1}^{p} R_j}$ and we recover the result in **Eq (6)**.

We infer the mean and risk averse reproduction numbers by combining estimates of the group reproduction numbers, $R_j$, generated from EpiFilter. We achieve this by sampling from the posterior distributions of these local estimates $\mathbf{P}(R_j(t)|I_1^T)$ to construct consensus posteriors $\mathbf{P}(X(t)|I_1^T)$ for $D$ and $E$ in a Monte Carlo manner according to **Eq (6)**. This involves computing the arithmetic and contraharmonic means of the samples at each time point. The contraharmonic mean is the ratio of the second to first raw moments of its inputs. We also use as a reference, the simple statistic max $R_j$, which involves taking maxima over the group samples. These consensus estimates underlie the plots in **Figs 3–6**.

Importantly, we observe that as both $D$ and $E$ are means, they have two key properties that define them as reproduction numbers. First, when all the local $R_j = a$ then $R = D = E = a$. Second, if we reduce transmissibility globally by $\frac{1}{a}$ (i.e., every $R_j$ is scaled by $\frac{1}{a}$, with $a > 1$) then all three statistics are also reduced by $\frac{1}{a}$. These properties ensure that our consensus statistics have the same interpretability as $R$. Specifically, $D$ and $E$ have a threshold around 1 and their estimated values reflect changes resulting from public health interventions, more transmissible variants (if instead the $R_j$ scale up by $a$) and population behaviours.

## Optimal experimental design with interconnected groups

Our framework above does not explicitly consider connections among the groups. Here we outline how our optimal designs can remain valid under models of realistic interconnections. Let $\rho_{x \to j}$ be the probability of an infection being introduced into a sink group $j$ from source group $x$ as in [43] with $\rho_{x \to x}$ as the probability of remaining within the source group. For $p$ groups, the renewal process that describes the incidence of new infections in group $j$ is $I_j \sim \mathbf{Pois}(\sum_{x=1}^{p} \rho_{x \to j} \Lambda_x R_x)$, ignoring explicit time indices. Consequently, $I_j$ contains information about the $R_x$. If we assume that introductions have the infectiousness of their source group and that we know the source group of the introductions, then the informative component of $I_j$ is then $I_j|R_x \sim \mathbf{Pois}(\rho_{x \to j} \Lambda_x R_x)$ (via a Bernoulli thinning of Poisson distributions).

Using earlier results, the Fisher information that $I_j$ contains about $R_x$ is $\rho_{x \to j} \Lambda_x R_x^{-1}$. We may collect the information about $R_x$ available from all the infection data by summing these terms as $\sum_{j=1}^{p} I_j|R_x \sim \mathbf{Pois}(\Lambda_x R_x)$. This follows from the infinite divisibility property of the Poisson formulation and as $\sum_{j=1}^{m} \rho_{x \to j} = 1$. Consequently, the total Fisher information about $R_x$ from all the incidence data is $\mathbf{FI}[R_x] = \Lambda_x R_x^{-1}$. This yields the same Fisher matrix as in **Eq (4)**. Consequently, our D- and E-optimal designs and other results are unchanged and valid under this formulation, which assumes that introductions have the infectiousness of their source group. This assumption holds for example when transmission heterogeneity arises from regional pathogenic variants, with variants forming groups with distinct $R_x$.

The converse, where introductions have the reproduction number of their sink group leads to $I_j \sim \mathbf{Pois}(\sum_{x=1}^{p} \rho_{x \to j} \Lambda_j R_j)$. If we let $\sum_{x=1}^{p} \rho_{x \to j} = a_j$, we get a diagonal Fisher matrix but now with terms $\mathbf{FI}[R_x] = a_x \Lambda_x R_x^{-1}$. This fits our framework in (see **Eq (10)**) if we constrain the sum of all the $a_x \Lambda_x = \alpha_x$ to still be $\Lambda$. Sink-based reproduction numbers may occur, for example, if groups demarcate areas with different population density or contact patterns and have distinct $R_x$ (that is acquired on entering that group). Both source and sink assumptions require

knowledge of the introductions or their $\rho_{x \to j}$ values. When the $\rho_{x \to j}$ are unknown or cannot be estimated, an alternative is to treat the introductions as input data as in [36].

This requires that we redefine the total infectiousness of group $j$ as $\Lambda_j = \sum_{u=1}^{s-1} w(u) \times (I_j(t-u) + M_j(t-u))$ with $I_j$ as the local infections of group $j$ and $M_j$ counting introductions into that group. As $\Lambda_j$ is treated as known we do not depart the framework of the main text and we recover our optimal designs (albeit with this redefined $\Lambda_j$). This convergence of results emerges because once we can ascertain the source and sink of infections, we can correctly assign them to their respective $R_x$ and construct a diagonal Fisher matrix. Other models of interconnectivity which instead propose inter-group reproduction numbers (e.g., [73]) do not directly fit our framework or possess non-diagonal Fisher information matrices and can be over-parametrised or non-identifiable without additional data or constraints.

## Supporting information

**S1 Appendix. This provides mathematical details on real-time estimates of reproduction numbers and on general risk averse properties of $E$.** Additionally it contains **Figs A-C**. **Fig A**: Real-time analysis of COVID-19 Delta strain dynamics in Israel. We repeat the analysis from **Fig 5** but truncate to key points in the epidemic time series to show real-time or prospective estimates of transmissibility using data up to those truncation points only. **Fig B:** Risk averse reproduction numbers for COVID-19 in Norway. We perform a similar analysis to **Fig 5** but for COVID-19 waves in Norway and compare transmissibility estimates to intervention times. **Fig C:** Incidence curves for 6 empirical COVID-19 datasets. We plot new infections for every dataset analysed in **Figs 6 and 7**, showing the epidemic curves by group and in total. (PDF)

## Author Contributions

**Conceptualization:** Kris V. Parag, Uri Obolski.

**Data curation:** Uri Obolski.

**Formal analysis:** Kris V. Parag.

**Funding acquisition:** Kris V. Parag, Uri Obolski.

**Investigation:** Kris V. Parag.

**Methodology:** Kris V. Parag.

**Project administration:** Kris V. Parag, Uri Obolski.

**Software:** Kris V. Parag.

**Validation:** Kris V. Parag, Uri Obolski.

**Visualization:** Kris V. Parag, Uri Obolski.

**Writing – original draft:** Kris V. Parag.

**Writing – review & editing:** Kris V. Parag, Uri Obolski.

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
