## [Decision Letter · Decision Letter 0]

24 Feb 2023

Dear Dr Parag,

Thank you very much for submitting your manuscript "Risk averse reproduction numbers improve resurgence detection" for consideration at PLOS Computational Biology.

As with all papers reviewed by the journal, your manuscript was reviewed by members of the editorial board and by two independent reviewers. In light of the reviews (below this email), we would like to invite the resubmission of a significantly-revised version that takes into account the reviewers' comments. In particular, referee 2 provides a number of comments on ways to strengthen the case for the effectiveness of the proposed quantity, and referee 1 raises relevant points for discussion, including the inter-connectedness of sub-populations.

We cannot make any decision about publication until we have seen the revised manuscript and your response to the reviewers' comments. Your revised manuscript is also likely to be sent to reviewers for further evaluation.

Sincerely,

Mercedes Pascual

Academic Editor

PLOS Computational Biology

Thomas Leitner

Section Editor

PLOS Computational Biology

Reviewer's Responses to Questions

**Comments to the Authors:**

Reviewer #1: The manuscript by Drs. Parag and Obolski deals with the analysis of weighting schemes aimed at improving the performance of the effective reproduction number (R) as a means of detecting resurgent epidemic dynamics. The authors' basic idea is that current approaches to estimate R from surveillance data tend to overplay the role of larger subpopulations, possibly leading to delays in identifying important signals from smaller subpopulations where transmission rebounds might actually start. To circumvent this limitation, the authors apply some information-theoretic principles from experimental design theory to present two alternative weighting schemes, one based on D-optimality (which, for the problem at hand, reduces to equal weighting for the effective reproduction numbers of the various subpopulations, independently of the active infections circulating therein) and the other based on E-optimality (in which the reproduction number of each subpopulation is weighted according to the fraction of the total reproduction number sum attributable to that subpopulation), respectively. Using numerical experiments of increasing complexity (from a toy model to a realistic application based on actual data), they propose that E-optimal reproduction numbers can produce an effective bias-variance trade-off.

The topic of the manuscript is undoubtedly interesting, as R is widely used in decision-making, despite its known limitations---one of which in fact being the underlying homogeneity assumption often made when estimating it from data. The analysis is well conducted and quite effectively explained, although the nitty-gritty of the methodology might be better understood by readers with some prior exposure to information theory. The manuscript is sufficiently well written and organized. Specific comments follow.

Main comments

- 2nd page, 2nd paragrapg: The homogeneity assumption is often done when trying to reconstruct R from data; this is not always the case, instead, when the estimation of R is based on other approaches, such as compartmental models calibrated with surveillance data. This family of models certainly has its own share of problems and limitations, but can in principle accommodate heterogeneous populations with a computational cost that is typically negligible compared to that of individual-based approaches. A comment on this might help put the initial discussion on the benefits and limitations of R in a broader perspective.

- 4th page, 2nd paragraph: I think that neglecting inter-group infections may perhaps be the most important limitation of this work, which expands (and likely improves) over previous homogeneous approaches by introducing a principled weighting of the different subpopulations, but does (understandably) little to account for the epidemiological interrelationships among them---or with the "outside world". The redistribution of information described in equation (4) is said to be able to account for "migrations" (a term that I suggest replacing with something less spatially and temporally connotated). However, the cited reference [28] makes explicit use of mobility fluxes among different subpopulations, the mapping of which onto the modeling framework presented in this manuscript does not appear completely straightforward. Moreover, the other reference cited in this context [21] makes an explicit distinction between local and imported cases, but it is not immediately clear how to introduce this feature in the alpha factors of equation (4) either. I believe that providing more information on how the proposed method can accommodate (or be extended to accommodate) such mixing/import dynamics would represent an important contribution toward a wider applicability of the method.

- page 12, fig 3 caption: "but overamplifies noise" is there a quantitative way to qualify this statement? I think it could indeed help explore some of the trade-offs involved in the selection of a weighting scheme for a consensus reproduction number.

Other comments

- page 3, 2nd paragraph: "it optimises... estimate uncertainty" unclear, please explain

- page 8, 3rd paragraph: "Both D and E... each group" unclear, please explain

- page 12, fig 3 caption: "do not signal control" quite vague, please rephrase

- page 12, 1st paragraph: "every group... is controlled", "being driven towards elimination" quite vague too, please rephrase (groups are likely neither controlled nor eliminated, disease transmission is)

- page 14, 3rd paragraph: "These are reasonable" what: hypotheses/assumptions/...?

- page 18, equation (7): are the age-of-infection indexes correct in the right-hand side expression, or are they switched?

- page 20, first paragraph: algorithm [49], to derive -> algorithm [49] to derive

Reviewer #2: In this study, the authors propose an alternative measure to the standard effective reproduction number (R), the risk averse reproduction number (E). The new measure aims at addressing some issues with the standard definition of R such as the low sensitivity to local resurges which might lead to a delayed detection of outbreak resurgence and therefore have important ramifications for policy responses.

Even though the proposed metric is interesting, the authors should try to refine/revise the manuscript to make a more compelling case of its usefulness. In the following I will provide a few comments to address this concern.

First, from the title and abstract of this paper, the stated goal of the authors is to promote the new measure E. However, in the remainder of the paper and especially in sections “D and E optimal reproduction numbers and their properties”, “Risk averse reproduction numbers are more representative of key dynamics”, and “Empirical application to COVID-19 across 20 cities in Israel” the authors actually focus on two “risk averse reproduction numbers”. This takes the reader’s focus away from their proposed main metric (E) which is the one discussed at the beginning and end of the paper. For this reason, I would recommend the authors to either move the discussion about D in the supporting information or to broaden their statements at the beginning and end of the paper to promote (more in general) “optimal-design/risk averse reproduction numbers” (i.e. both D and E).

Second, the “synthetic examples” showcase situations in which E might be better at predicting resurgence than R, however it is not clear whether those simulations come from an “organic model” or rather from independent simulations of local outbreaks which are then simply assumed to be part of the same (global) geography. Let me clarify this point. If one of the main arguments brought forward is that E is better than R at detecting a local resurgence that might become a global resurgence, then the authors should provide synthetic simulations in which - for example - they use a metapopulation model showing that a local resurgence in a context of connected subpopulation will transform into a global resurgence because of the delayed R-dynamics (vs E-dynamics) that will prevent a timely policy response.

Third, related to the point above, it would be useful if the authors were to show the “policy response” in action using R vs E (for example in synthetic simulations). In the example of the metapopulation model above, this would entail running simulations in which - on the one hand - the timing of the policy interventions is driven by changes in R and - on the other hand - a scenario in which - instead - the timing is dictated by changes in E. This would provide a way to quantify the policy impact of using E vs R. Furthermore, this could also help identifying cases in which even E might be moving “too slowly” to prevent local resurgences to become global problems (i.e. are there cases in which the surges are so rapid that both R and E, especially when accounting for delayed reporting of surveillance data, fail to provide enough lead times for policy makers to react?).

Fourth, in the section “Empirical application to COVID-19 across 20 cities in Israel” the authors show that using E might have helped in deciding to start the booster campaign earlier in time. Could the authors quantify this? For example by comparing the dates in which E estimates were above 1 vs when R estimates were above 1 (considering also the role played by the uncertainty of such estimates).

Fifth, in the Discussion, the authors identify one of the possible Achille’s heel in promoting the use of E vs R: the risk averse reproduction number requires infection time and incidence data (or proxies of them) at the local resolution. The authors do not seem to make a strong case in addressing this weakness rather than a general concession to the fact that “real-time decision-making depends on quality of data”. As an addition to that statement, it would be useful if the authors could show more than one real world example in which they applied their proposed metric. In other words, can the authors provide additional case studies (like the one provided for Israel) showing that indeed – for example in the case of the “variants waves” of COVID-19 – this approach could have been used in different countries to fine tune the timing of the booster campaigns (or of other interventions)?

In other words, one of the main weaknesses I see with this manuscript in its current form is that it does not make a strong compelling case for real world applications of the proposed E measure, even though that seems to be the biggest selling point of this novel metric. Therefore, I would recommend the authors to improve either the “synthetic simulations” part of the manuscript by using more realistic scenarios or/and by adding additional case studies besides the ones they already consider.

**Have the authors made all data and (if applicable) computational code underlying the findings in their manuscript fully available?**

Reviewer #1: Yes

Reviewer #2: Yes

PLOS authors have the option to publish the peer review history of their article (what does this mean?). If published, this will include your full peer review and any attached files.

Reviewer #1: No

Reviewer #2: No
---

## [Decision Letter · Decision Letter 1]

18 May 2023

Dear Dr Parag,

Thank you very much for submitting your manuscript "Risk averse reproduction numbers improve resurgence detection" for consideration at PLOS Computational Biology. As with all papers reviewed by the journal, your manuscript was reviewed by members of the editorial board and by the two independent reviewers who had provided comments on your original submission.

Based on the reviews, we are likely to accept this manuscript for publication, providing that you modify the manuscript to revise and improve the section on ABMs vs. network and compartmental models as indicated by Reviewer 2.

Sincerely,

Mercedes Pascual

Academic Editor

PLOS Computational Biology

Lucy Houghton

Staff

PLOS Computational Biology

Reviewer's Responses to Questions

**Comments to the Authors:**

Reviewer #1: I have now read the revised version of the manuscript by Drs. Parag and Obolski. I think that the authors have done a good job of answering the reviewers' comments and revising their manuscript accordingly. I appreciate the wider context provided in the introduction, as well as the extensive methodological discussion on how the mathematical framework can accommodate interconnected populations and/or case imports. Last but not least, the stronger focus on the E metric and the inclusion of a wider set of real-world applications represent, in my view, compelling additions to an already interesting contribution. From my side, I do not have any further comments on this paper.

Reviewer #2: The authors answered and clarified all main concerns of this reviewer.

As for the new additions to the paper, I would like to kindly ask the authors to improve/revise the part of the manuscript in which they discuss agent-based models (ABMs) vs network models vs compartmental models.

"""This is a great point. We include a new paragraph in the Introduction on compartmental models discussing how they can account for heterogeneous transmission characteristics e.g., by including multiple infectious classes. While these do not possess the computational expense of network or agent-based models (which incorporate heterogeneities explicitly) they do require knowledge of increasing numbers of parameters and distributions as their complexity (and descriptive power) increase. We distinguish these from the renewal and autoregressive models that are commonly used for real-time estimation and forecasting, and which underlie our investigation. Hopefully this better contextualises our results."""

While this new section certainly provides some additional clarity, it is somehow still a bit confusing.

For example, distinguishing between ABM and network models is quite an arbitrary distinction as individual-based network models are ABMs; and all agent-based models can be also seen through the lens of being network models. Furthermore, the citations provided by the authors seem quite outdated. A lot of work has been done in this space in the last 10/15 years.

If it is certainly true that agent-based models require high-resolution data (to run/parametrize the model, not necessarily to fit them), recent years have shown how the availability of - for example - large-scale mobility datasets has enabled the creation of very detailed ABM models (see for example recent works by Alberto Aleta). Furthermore, it has been shown that ABMs can be used to create stratified contact matrices (e.g. by age, see Mistry, D., et al. Inferring high-resolution human mixing patterns for disease modeling. Nat Commun 12, 323, 2021) which can in turn be used to introduce heterogeneous mixing in more traditional compartmental modeling approaches (including metapopulation models).

In addition, network models do not necessarily need detailed contact tracing data as contact networks can be approximated either by using contact matrices (built via ABM sims or survey data) or by using high-resolution mobility data (e.g. SafeGraph, Cuebiq, etc..) which can be used to map individual behaviors (e.g. visitation to different POIs) during an average day, therefore allowing for the creation of “probabilistic” contact networks (see A. Aleta works).

Lastly, also when discussing metapopulation models it would be useful to update the citations as a lot of work has been done in recent years, including a lot of contributions during the recent COVID-19 pandemic (e.g. see recent works by Lauren Meyers’ group, Jeffrey Shaman’s, Alessandro Vespignani’s group, etc..).

To put it differently, this new section is a bit “rough around the edges” and it would benefit from another round of edits. Furthermore, I would like to point out that the suggested citations/authors listed above are merely examples, the main point I am trying to put across is that this section still needs some work.

Minor comment:

In Fig 6 and 7, it would be useful to have the color legend for the 3 curves also in the main figure (as done in the other figures), rather than explaining the color scheme of the lines only in the figure caption.

**Have the authors made all data and (if applicable) computational code underlying the findings in their manuscript fully available?**

Reviewer #1: Yes

Reviewer #2: Yes

PLOS authors have the option to publish the peer review history of their article (what does this mean?). If published, this will include your full peer review and any attached files.

Reviewer #1: No

Reviewer #2: No

Figure Files:

Data Requirements:

Reproducibility:

References:

---

## [Editor Report · Decision Letter 2]

6 Jul 2023

Dear Dr Parag,

We are pleased to inform you that your manuscript 'Risk averse reproduction numbers improve resurgence detection' has been provisionally accepted for publication in PLOS Computational Biology.

Best regards,

Mercedes Pascual

Academic Editor

PLOS Computational Biology

Thomas Leitner

Section Editor

PLOS Computational Biology

---

## [Editor Report · Acceptance letter]

15 Jul 2023

PCOMPBIOL-D-22-01819R2 

Risk averse reproduction numbers improve resurgence detection

Dear Dr Parag,

I am pleased to inform you that your manuscript has been formally accepted for publication in PLOS Computational Biology. Your manuscript is now with our production department and you will be notified of the publication date in due course.

With kind regards,

Zsofia Freund
